# Therapeutic Efficacy of Curcumin Enhanced by Microscale Discoidal Polymeric Particles in a Murine Asthma Model

**DOI:** 10.3390/pharmaceutics12080739

**Published:** 2020-08-06

**Authors:** Jun Young Park, Ga Eul Chu, Sanghyo Park, Chaewon Park, Susmita Aryal, Won Jun Kang, Won Gil Cho, Jaehong Key

**Affiliations:** 1Department of Nuclear Medicine, Severance Hospital, Yonsei University College of Medicine, 50-1 Yonsei-ro, Seodaemun-gu, Seoul 03722, Korea; abies60@naver.com (J.Y.P.); mdkwj@yuhs.ac (W.J.K.); 2Department of Anatomy, Yonsei University Wonju College of Medicine, 20 Ilsan-ro, Wonju 26426, Korea; xiahgaeul@naver.com; 3Department of Biomedical Engineering, Yonsei University, Gangwon-do, Wonju 26493, Korea; hapy9490@yonsei.ac.kr (S.P.); chaewon_p@yonsei.ac.kr (C.P.); sabica_nymph92@yahoo.com (S.A.)

**Keywords:** discoidal polymeric particle, asthma, drug delivery system, microparticle, curcumin

## Abstract

Curcumin is considered a potential anti-asthmatic agent owing to its anti-inflammatory properties. The objective of the present study was to prepare curcumin-containing poly(lactic-*co*-glycolic acid)-based microscale discoidal polymeric particles (Cur-PLGA-DPPs) and evaluate their anti-asthmatic properties using a murine asthma model. Cur-PLGA-DPPs were prepared using a top-down fabrication method. The prepared Cur-PLGA-DPPs had a mean particle size of 2.5 ± 0.4 μm and a zeta potential value of −34.6 ± 4.8 mV. Ex vivo biodistribution results showed that the Cur-PLGA-DPPs mainly accumulated in the lungs and liver after intravenous injection. Treatment with Cur-PLGA-DPPs effectively suppressed the infiltration of inflammatory cells in bronchoalveolar lavage fluid, and reduced bronchial wall thickening and goblet-cell hyperplasia compared to those in the phosphate-buffered-saline-treated control group. No significant changes in hematology and blood biochemistry parameters were observed after treatment with Cur-PLGA-DPPs. At equal curcumin concentrations, treatment with Cur-PLGA-DPPs exhibited better therapeutic efficacy than treatment with free curcumin. Our results suggest that the microscale Cur-PLGA-DPPs can be potentially used as a lung-targeted asthma therapy.

## 1. Introduction

Asthma is a chronic inflammatory disease of the lungs that is characterized by reversible airway obstruction, airway hyper-responsiveness, and airway inflammation [1,2]. Chronic inflammation of the airways typically results in the activation and infiltration of inflammatory cells, including eosinophils, mast cells, macrophages/monocytes, neutrophils, and T lymphocytes [3]. Inflammation is an essential protective function that serves to counteract pathogen invasion and clear damaged cells and toxic compounds [4]. However, inappropriate immune activation may elicit dysregulation of inflammatory mechanisms, which can induce an overproduction of cytokines by inflammatory cells. Asthma is most commonly treated with corticosteroids, which suppress the production of inflammatory mediators and prevents the recruitment or activation of inflammatory cells [5]. Despite the excellent therapeutic effects of corticosteroids, their practical application is limited owing to considerable local and systemic side effects associated with long-term use. These side effects include dysphonia, reflex cough, bronchospasm, suppression of the hypothalamic–pituitary–adrenal axis, reduced bone mineral density, skin thinning and bruising, osteoporosis, growth retardation, cataracts, glaucoma, and Cushing’s syndrome [6,7].

Curcumin is a natural polyphenol extracted from the rhizomes of turmeric (*Curcuma longa* Linn.). Previous studies have demonstrated its antioxidant, antimicrobial, anti-inflammatory, and anticarcinogenic properties. These functions are achieved via the modulation of transcriptional factors, inflammatory cytokines, protein kinase, enzymes, signaling receptors, growth factors, cell-surface adhesion molecules, and anti-apoptotic proteins [8,9]. Curcumin is considered a potent anti-asthmatic agent owing to its anti-inflammatory properties [10]. It attenuates airway inflammation and hyper-responsiveness via the inhibition of the nuclear factor-kappa B pathway and its nitric-oxide-regulating effects [11,12]. Clinical trials have demonstrated that curcumin is safe, well-tolerated, and nontoxic, even at high doses [13]. Despite its apparent efficacy and safety, the clinical application of curcumin is limited due to the poor bioavailability caused by its extremely low aqueous solubility, low intrinsic activity, low absorption rate, fast metabolism, and rapid clearance [14].

Various drug-delivery systems, including liposomes, nanoparticles, microemulsions, cyclodextrin complexes, solid dispersions, and micelles, have been tested to improve the in vivo therapeutic applications of curcumin [15,16,17,18]. Among these systems, poly(lactic-*co*-glycolic acid) (PLGA) particles are the most commonly applied vehicles for drug delivery, owing to their biodegradability and biocompatibility [19]. Previous studies have shown that the bioavailability of curcumin improves when it is administered via curcumin-loaded PLGA nanoparticles, compared to the administration of free curcumin [20,21].

The size and shape of particles can affect their biological behavior in vivo [22,23]. Decuzzi et al. demonstrated that discoidal particles accumulate in significantly higher concentrations in the lungs than spherical particles or quasi-hemispherical particles [24]. This study performed by them also showed that 3.0 μm sized particles exhibited significantly higher lung uptake than the 0.7, 1.0, and 2.5 μm sized particles. In addition, in our recent in vivo biodistribution and bioimaging study on 3.0 μm sized discoidal particles, we found that 3 μm sized discoidal particles were mainly accumulated in lung parenchyma and remained there for up to three days after intravenous administration [25]. The results of these studies suggested the potential of micro-ized discoidal particles for drug delivery to the lungs. In the present study, PLGA-based microscale discoidal polymeric particles (DPPs) containing curcumin (termed Cur-PLGA-DPPs) were prepared using a top-down fabrication method. The physicochemical characteristics of these Cur-PLGA-DPPs, including morphology, particle size, zeta potential, drug-loading efficiency, and in vitro curcumin-release behavior, were investigated. Furthermore, the therapeutic efficacy of Cur-PLGA-DPPs in a murine asthma model was examined.

## 2. Materials and Methods

### 2.1. Materials

All reagents were commercially available and were used as purchased. Poly(D,L-lactide-*co*-glycolide) ((PLGA); Resomer^®^ RG503H, acid terminated, molecular weight: 24–38 kDa, lactide:glycolide ratio = 50:50), poly(vinyl alcohol) (PVA) (80% hydrolyzed, molecular weight: 9–10 kDa), ovalbumin ([OVA]; grade V), and curcumin (≥95% purity, Cur) were purchased from Sigma-Aldrich (St. Louis, MO, USA). Polydimethylsiloxane (PDMS) (Sylgard 184 Silicone Elastomer Kit) was purchased from Dow Corning (Midland, MI, USA). Cy7-labeled PLGA (molecular weight: 15–35 kDa, lactide:glycolide ratio = 50:50) was obtained from RuiXi Biological Technology (Xi’an, China).

### 2.2. Preparation of Curcumin-Loaded PLGA-DPPs

Microsized Cur-PLGA-DPPs were prepared using a top-down fabrication method, as described previously [25,26]. Briefly, a silicon master template, which had fixed cylindrical holes, with a diameter of 3 μm and a height of 1.49 μm, was constructed using electron-beam lithography. To prepare a PDMS template, PDMS and elastomer were mixed at a ratio of 10:1 and spread onto the silicon master template. The PDMS mixture was dried at 60 °C for 6 h and then peeled off the silicon master template. To prepare the PVA template, we applied 6% (*w/v*) aqueous PVA solution to the surface of the PDMS template. Subsequently, it was dried at 30 °C and peeled off after 15 h. PVA templates showed the same patterns as the silicon master template. PLGA (60 mg) was dissolved in 100 μL of dichloromethane–chloroform (1:1, *v/v*). To this solution, 60 mg of curcumin (dissolved in 100 μL DMSO) was added, followed by stirring to ensure homogenization. The curcumin–PLGA mixture was evenly applied to the wells on the PVA template using a razor and then dried at room temperature for 5 min. The PVA template was dissolved in deionized water for 3 h, and the solution was passed through a 100 µm nylon filter to remove large debris. The pellets were collected by centrifugation at 1000× *g* for 2 min.

### 2.3. Characterization of Cur-PLGA-DPPs

Particle size distribution was measured using a Multi-Sizer Coulter Counter (Beckman Coulter Inc., Miami, FL, USA). The zeta potential and particle size of the Cur-PLGA-DPPs were determined using a Zetasizer Nano ZS90 (Malvern Instruments Ltd., Malvern, UK) instrument after dilution with deionized water (1 mg/mL). The surface morphology of the Cur-PLGA-DPPs was examined using a JEOL-7800F field emission scanning electron microscope (JEOL, Tokyo, Japan), with an accelerating voltage of 0.8 kV. Fluorescence images of Cur-PLGA-DPPs were acquired using a Zeiss LSM-700 confocal laser scanning microscope (Carl Zeiss, Oberkochen, Germany).

### 2.4. Drug Loading and In Vitro Drug Release Test

Drug loading (percent) and release of Cur-PLGA-DPPs were measured using a Synergy™ HTX multimode microplate reader (Bio-Tek Instruments, Vermont, VT, USA). Before measurement, a standard calibration curve of curcumin was prepared using six standard solutions (7.8–250 μg/mL). To measure loading efficiency, 1 mg of lyophilized curcumin DPPs was dissolved in 1 mL DMSO. The suspension was vigorously vortexed, and the absorbance of the mixture was measured at 420 nm. The percentage of drug loading (%DL) was determined using the following formula:%DL = [Amount of the drug in particles/Amount of particles] × 100(1)

Drug release of Cur-PLGA-DPPs was determined using a dialysis method in phosphate-buffered saline ((PBS); pH 5.5 and pH 7.4) containing 2% Tween-80. The pH of the PBS solution was adjusted to 5.5 using 0.1 M acetate buffer. Tween-80 was added to the PBS solution to counteract the low aqueous solubility of curcumin. One milligram of lyophilized Cur-PLGA-DPPs, dissolved in 1 mL PBS, was placed in a 10 kDa cut-off dialysis membrane tube (Spectrum™ Spectra/Por™ 7, Spectrum Labs, Rancho Dominquez, CA, USA). The bag was immersed in 10 mL PBS containing 2% Tween-80 and was kept at 37 °C under constant stirring at 140 rpm. At designated time intervals, 100 μL samples were removed from the incubation medium, and the same volume of PBS solution was added. The amount of curcumin released from Cur-PLGA-DPPs was analyzed using a multimode microplate reader, as detailed above.

### 2.5. In Vitro Stability

The stability of Cur-PLGA-DPPs in 50% (*v/v*) mouse serum at 37 °C was assessed for up to 72 h. As a control, the in vitro stability of Cur-PLGA-DPPs in PBS (pH 7.4) was recorded. Samples were collected at 0.5, 1, 3, 6, 12, 24, 48, and 72 h, and the concentration of Cur-PLGA-DPPs was monitored using a Multi-Sizer Coulter Counter (Beckman Coulter Inc.).

The potential effects of pH on Cur-PLGA-DPP stability were examined using different pH conditions. Cur-PLGA-DPPs were incubated at different pH values (pH 5, 6, 7, and 8) for up to 2 h. Particle size and distribution were measured by dynamic laser light scattering (DLS) at 25 °C using a Zetasizer Nano ZS90 (Malvern Instruments Ltd.) instrument.

### 2.6. Murine Asthma Model

Female Balb/c mice, aged 6–8 weeks, were obtained from Orient Bio Inc. (Gyeonggido, South Korea) and kept under standard conditions with air filtration. The murine asthma model was established using ovalbumin (OVA), as reported previously, with minor modifications [27]. Briefly, Balb/c mice were sensitized by intraperitoneal injection of OVA (50 μg) in a suspension of 4 mg aluminum hydroxide on Days 0 and 7. Starting from Day 18, the mice were intranasally challenged with 2% OVA in saline solution for three consecutive days. Control mice were sensitized and challenged with 0.9% NaCl. All animal experiments were approved by the Institutional Animal Care and Ethics Committee of the Yonsei Laboratory Animal Research Center.

### 2.7. Ex Vivo Imaging

On Day 21, OVA-induced asthmatic mice were injected in the tail vein with 500 μg Cy7-labeled Cur-PLGA-DPPs and were sacrificed at 1 h, 3 h, 6 h, 1 day, 3 days, 5 days, and 7 days (three mice at each time point) after injection. Liver, muscle, spleen, heart, lung, and kidney tissues were collected, and ex vivo imaging was performed using the in vivo Imaging System (IVIS) Spectrum (Perkin Elmer, Santa Clara, CA, USA). Images were analyzed using the IVIS imaging software (Perkin Elmer). OVA-induced asthmatic mice were fed an alfalfa-free diet for 14 days before bioluminescence imaging to reduce intestinal autofluorescence.

### 2.8. In Vivo Therapeutic Efficacy

Mice were assigned to OVA-induced asthma groups with different treatments and a PBS-treated control group. OVA-induced asthmatic mice were randomly assigned to three treatment groups (four to five animals per group): (i) an OVA group, (ii) an OVA + Cur-PLGA-DPP group (at a dose of 5 or 25 mg/kg), and (iii) an OVA + curcumin group (4 mg/kg, which was equivalent to the curcumin concentration in 25 mg/kg Cur-PLGA-DPPs). Starting from Day 21, the animals received weekly injections of 0.9% NaCl, Cur-PLGA-DPPs, or curcumin via the tail vein, for six weeks. The mice in the control and OVA groups were treated with the corresponding volumes of 0.9% NaCl solution. After 1 h, all OVA-induced asthmatic mice received 1% OVA intranasally to maintain sensitivity.

### 2.9. In Vivo Toxicity

After the six week treatment period, blood was collected via cardiac puncture and was placed in Microtainer^®^ EDTA-coated tubes and serum separator tubes (BD Biosciences, Franklin Lakes, NJ, USA). Hematological assays were performed using the HEMAVET^®^ 950 hematology system (Drew Scientific, Waterbury, CT, USA). Serum levels of aspartate aminotransferase (AST), alkaline phosphatase (ALP), alanine aminotransferase (ALT), creatinine (CRE), and blood urea nitrogen (BUN) were measured using an automated biochemistry analyzer (Fuji Dri-chem 4000i, Fujifilm Corp., Tokyo, Japan).

### 2.10. Bronchoalveolar Lavage Fluid Collection and Cell Counts

After six weeks of Cur-PLGA-DPP treatment, bronchoalveolar lavage fluid (BALF) was collected as described in a previous study, with a minor modification [28]. BALF was collected by cannulating the trachea using a 22 gauge intravenous catheter (BD Biosciences, San Jose, CA, USA), followed by flushing the lungs with 1 mL cold PBS (pH 7.4). BALF was centrifuged at 800× *g* at 4 °C for 10 min. Cell pellets were resuspended in PBS, and the number of nucleated cells was counted using a hematocytometer. To determine the differential cell count, BALF was centrifuged at 800 rpm for 3 min using a Cytospin cytocentrifuge^®^ 3 (Thermo Scientific, Waltham, MA, USA). Slides were stained using a Diff-Quik^®^ staining kit (Sysmex, Kobe, Japan). In total, 400 nucleated cells per slide were examined, in a minimum of two different fields of view, using an Olympus BH-2 microscope (Olympus, Tokyo, Japan).

### 2.11. Histopathology

After the Cur-PLGA-DPP treatment, the left lung of each mouse was harvested and was then fixed using 4% paraformaldehyde, embedded in paraffin, and sectioned at 5 μm thickness. Lung sections were stained using hematoxylin and eosin (H&E) and periodic acid-Schiff (PAS), using standard histological methods. Stained histology slides were scanned using an Aperio AT2 Digital Whole Slide scanner (Leica Microsystems GmbH, Wetzlar, Germany).

### 2.12. Statistical Analyses

Data are shown as the mean ± standard deviation. Statistical analysis was performed using the GraphPad Prism 7 software (San Diego, CA, USA). Treatment effects were tested using one-way analysis of variance and Student’s *t*-test. Statistical significance is reported at *p* < 0.05.

## 3. Results and Discussion

### 3.1. Preparation and Characterization of Cur-PLGA-DPPs

#### 3.1.1. Size and Zeta Potential

The mean sizes of PLGA-DPPs and Cur-PLGA-DPPs were 2.5 ± 0.7 μm and 2.5 ± 0.4 μm, respectively, and their zeta potential values were −27.7 ± 4.7 mV and −34.6 ± 4.8 mV, respectively (Table 1). In general, PLGA-based nanoparticles exhibited a negative zeta potential because of the free carboxylic terminal groups of PLGA. However, the surface charge of Cur-PLGA-DPPs decreased after the addition of curcumin. This change may be attributed to the negatively charged hydroxyl groups of curcumin. Previous studies showed that the zeta potential value of curcumin in water is approximately −40 mV [29].

#### 3.1.2. Surface Morphology

A scanning electron micrograph of Cur-PLGA-DPPs revealed that the produced Cur-PLGA-DPPs were discoidal and had a smooth surface (Figure 1a). The morphology of Cur-PLGA-DPPs showed no apparent differences from that of PLGA-DPPs [25]. Thus, the addition of curcumin did not affect particle size and morphology. Curcumin is of bright yellow coloration, and it is naturally fluorescent. Cur-PLGA-DPPs also appeared yellow upon bright-field microscopy (Figure 1b). Moreover, fluorescence microscopy images showed that Cur-PLGA-DPPs emitted green fluorescence when exposed to UV radiation (Figure 1c).

#### 3.1.3. Loading Capacity and Release Profiles

The drug-loading efficiency in Cur-PLGA-DPPs of the optimized batch was 16.3 ± 0.1% (Table 1). In vitro release of curcumin from Cur-PLGA-DPPs was assessed under neutral and acidic pH conditions (Figure 2a), and the release profiles showed sustained release phases under both the conditions. At pH 7.4, Cur-PLGA-DPPs exhibited a low burst effect, with 6.9 ± 0.5% curcumin released within the first two days. However, under acidic pH conditions, 16.0 ± 0.4% of the encapsulated curcumin was released over the same time duration. After six days, the release rates of curcumin were 48.4 ± 1.6% at pH 7.4 and 83.9 ± 2.7% at pH 5.5. The drug-release profiles of Cur-PLGA-DPPs suggest a faster release of curcumin under acidic conditions than at neutral pH. This change may be explained by the faster degradation of the PLGA polymers at acidic than at neutral pH [30]. PLGA polymers are degraded via to the hydrolysis of the ester bond between lactic acid and glycolic acid [19], and hydrolytic degradation of PLGA can be accelerated under acidic conditions [31].

### 3.2. In Vitro Stability of Cur-PLGA-DPPs

The stability of Cur-PLGA-DPPs in mouse serum and PBS was assessed at different time points after particle incubation at 37 °C. Cur-PLGA-DPPs were present in PBS at 87.5 ± 0.9%, 50.1 ± 0.6%, and 45.5 ± 1.1% after incubation for 1, 24, and 72 h, respectively (Figure 2b). However, Cur-PLGA-DPPs degraded significantly faster (*p* < 0.001) in the mouse serum, with 51.6 ± 1.5% of Cur-PLGA-DPPs traceable in mouse serum after incubation for 3 h, and only 14.3 ± 0.7% traceable after incubation for 72 h. These in vitro stability data suggest that certain components of the mouse serum exerted a stronger effect on Cur-PLGA-DPPs than PBS.

Cur-PLGA-DPPs were incubated at different pH values to examine the potential effects of pH on stability. DLS was used to monitor the size of Cur-PLGA-DPPs at 25 °C for up to 2 h. Our results confirmed that Cur-PLGA-DPPs remained stable at pH 6, 7, and 8 (Figure 3). However, particle size decreased with increasing incubation time at pH 5.

### 3.3. Ex Vivo Imaging of Cy7-Cur-PLGA-DPPs

To investigate the ex vivo organ distribution of Cur-PLGA-DPPs, we injected OVA-induced asthmatic mice with Cy7-labeled Cur-PLGA-DPPs. Mice were sacrificed at predetermined time points, and their organs were imaged using an optical imaging system (Figure 4). Cy7-labeled Cur-PLGA-DPPs were found predominantly in the lungs, liver, and spleen. The strongest signal intensity was observed in lung tissue 1 h after injection, following which the fluorescence intensity decreased. Cy7-labeled Cur-PLGA-DPPs initially accumulated in the liver after 3–6 h; the fluorescence signal was still discernible in liver tissue up to three days after injection, whereas only weak signals were observed in the lung tissue at this time point. The uptake did not differ significantly between the muscle, heart, and kidney tissues. A previous study showed that one day after the injection of micro-sized PLGA-DPPs, their accumulation in the lung tissue was about three-fold higher than that in the liver tissue [25]. However, ex vivo imaging of Cy7-labeled Cur-PLGA-DPPs showed significantly higher fluorescence signals in the liver than in the lung at the same time points. Higher accumulation in the liver may be attributed to the size reduction of Cur-PLGA-DPPs during the course of degradation, considering that discoidal polymeric nanoconstructs, measuring ~1000 nm, accumulate in the liver [26]. Our ex vivo imaging data correlated with in vitro stability of Cur-PLGA-DPPs. Serum stability results suggested that in the mouse serum, half the amount of administered Cur-PLGA-DPPs degraded within the first 3 h.

### 3.4. In Vivo Cytotoxicity of Cur-PLGA-DPPs

To investigate the potential toxic effects of Cur-PLGA-DPPs, hematological and biochemical analyses were conducted on normal Balb/c mice. Changes in serum AST, ALP, ALT, CRE, and BUN levels were measured. Each group was treated with different doses of Cur-PLGA-DPPs (5 and 25 mg/kg) and curcumin (4 mg/kg) for six weeks, as shown in Table 2. Administration of Cur-PLGA-DPPs or curcumin did not exert significant effects on the biochemical parameters, including AST, ALP, ALT, CRE, and BUN, compared with the control. Treatment with Cur-PLGA-DPPs did not have any statistically significant effect on the hematological parameters (Appendix A). Our results, therefore, suggest that Cur-PLGA-DPP treatments caused no observable toxicity or side effects at doses up to 25 mg/kg.

### 3.5. In Vivo Therapeutic Efficacy of Cur-PLGA-DPPs

#### 3.5.1. BALF Analysis

The effects of Cur-PLGA-DPPs on the BALF total cell count and histological changes in the lung tissue were examined in OVA-induced asthmatic mice. Bronchoalveolar lavage is a useful method for monitoring inflammatory processes during various lung diseases [32]. Infiltration of inflammatory cells into the lungs is a typical histopathological characteristic of asthma [33]. The total number of BALF cells in the OVA-challenged treatment group was increased compared to that in the control group (Figure 5a,b). However, treatment with 5 mg/kg or 25 mg/kg Cur-PLGA-DPPs significantly decreased the abundance of inflammatory cells in BALF compared to that in the OVA group (*p* < 0.05). The total number of BALF cells in the OVA + 25 mg/kg Cur-PLGA-DPP-treated group was significantly lower than that in the OVA + 5 mg/kg Cur-PLGA-DPP-treated group (*p* < 0.05). Administration of 4 mg/kg curcumin also significantly reduced the total number of BALF cells, compared with that in the OVA group. However, asthmatic mice treated with OVA + 25 mg/kg Cur-PLGA-DPPs produced lower total cell numbers than mice in the 4 mg/kg curcumin-treated group (*p* < 0.05).

#### 3.5.2. Histological Evaluation

Histological staining with H&E was performed to assess the infiltration of inflammatory cells into the lung tissue of asthmatic mice. OVA-challenged mice showed extensive inflammatory cell infiltration, with prominent eosinophil infiltration in the peribronchial and perivascular areas. In addition, peribronchiolar smooth muscle hyperplasia and fibrosis were observed (Figure 6a). However, asthmatic mice treated with 25 mg/kg Cur-PLGA-DPPs exhibited a significant reduction in bronchial wall thickening and inflammatory cell infiltration in the peribronchial regions compared to the mice in the OVA group. The lungs of mice in the OVA + 25 mg/kg Cur-PLGA-DPP-treated group displayed less inflammatory cell infiltration and bronchial wall thickness than those of mice in the OVA + 5 mg/kg Cur-PLGA-DPP- and OVA + 4 mg/kg curcumin-treated groups. PAS staining was performed to identify goblet cells of the bronchial epithelium. The OVA group showed significant goblet-cell hyperplasia compared with the control group (Figure 6b). In contrast, administration of 5 mg/kg or 25 mg/kg Cur-PLGA-DPPs led to a reduction in the number of PAS-positive cells compared with the OVA group. The number of PAS-positive cells in the OVA + 25 mg/kg Cur-PLGA-DPP-treated group was lower than that in the OVA + 5 mg/kg Cur-PLGA-DPP- and 4 mg/kg curcumin-treated groups. These findings indicate that 25 mg/kg Cur-PLGA-DPPs had a more pronounced therapeutic effect than 4 mg/kg free curcumin.

To investigate the anti-inflammatory effects of blank PLGA-DPPs, OVA-induced asthmatic mice were injected with 25 mg/kg blank PLGA-DPPs. H&E staining of lung sections showed massive infiltration of inflammatory cells in the peribronchial and perivascular areas (Figure 7). PAS staining indicated a considerable number of purple-colored mucin-positive goblet cells in the bronchial epithelium. These findings suggest that blank PLGA-DPPs had no anti-inflammatory effect on the lung tissue of asthmatic mice.

The physical stability of polymeric particles is critical, because low stability can reduce drug bioavailability. In this study, in vitro serum stability and ex vivo organ distribution data indicated that the physical stability of Cur-PLGA-DPPs was relatively low. Various factors, including zeta potential, pH, particle size, ionic strength, molecular weight, and physicochemical properties, can affect the physical stability of polymeric particles [34,35,36]. Zeta potential is an important parameter that affects colloidal stability, because high repulsion forces prevent the aggregation of particles in solution. Particles with a zeta potential value above +30 mV or below −30 mV are generally considered stable [37]. In this study, the zeta potential value of Cur-PLGA-DPPs was −34.6 ± 4.8 mV, suggesting sufficient dispersion stability. In vitro stability studies also showed that Cur-PLGA-DPPs were stable between pH 6 and 8.

The loading amount of drugs and their chemical properties can also affect the physical stability of polymeric particles [19]. In the top-down fabrication method, polymers and drugs are loaded in a fixed volume in wells on a template. The amount of polymer is reduced as the drug is loaded, which can affect the physical stability of polymeric particles. However, this limitation of the top-down method can be overcome by altering the polymer’s composition. The biodegradation rate of PLGA is affected by the molar ratio of lactic acid (LA) and glycolic acid (GA) [19,31]. LA comprises a methyl group and is thus more hydrophobic than GA, and previous studies showed that PLGA with a higher content of LA degrades more slowly than PLGA containing more GA [30,38]. In this study, PLGA with a 50:50 ratio of LA and GA was used to produce Cur-PLGA-DPPs. PLGA with a higher LA content may decrease the degradation rate of polymeric particles. Moreover, curcumin is a hydrophobic molecule, and higher hydrophobicity of polymers can also increase the physical stability of polymeric particles, e.g., through hydrophobic interactions between curcumin and LA. Further studies are required to elucidate whether the physical stability of curcumin-loaded PLGA-DPPs depends on the ratio of LA to GA. Enhanced in vivo stability of Cur-PLGA-DPPs can lead to higher accumulation in the lung and improved therapeutic results.

A previous study suggested that microsized PLGA-DPPs are promising as a lung-targeted drug-delivery system [25]. In the current study, we found that 3 μm curcumin-containing PLGA-DPPs reduce inflammatory cell infiltration, bronchiolar wall thickening, and goblet-cell hyperplasia in a dose-dependent manner in the lungs of mice with OVA-challenged asthma. Furthermore, Cur-PLGA-DPPs showed a higher therapeutic efficacy than curcumin administered alone at the same dose, which may be attributed to enhanced curcumin bioavailability upon administration of curcumin as discoidal polymeric particles. The underlying pharmacokinetics were not examined in the present study. However, our findings suggest that Cur-PLGA-DPPs exert a better therapeutic effect for asthma than free curcumin.

## 4. Conclusions

In the present study, microscale Cur-PLGA-DPPs were designed as a polymeric vehicle of curcumin to enhance in vivo efficacy. Cur-PLGA-DPPs were successfully produced using a top-down fabrication method. The in vitro release experiment showed sustained release of curcumin from Cur-PLGA-DPPs, and ex vivo biodistribution experiments indicated that Cur-PLGA-DPPs accumulated mainly in the lungs, liver, and spleen. Administration of Cur-PLGA-DPPs effectively reduced the number of inflammatory cells in BALF and lung tissue, and also reduced the thickness of the bronchial wall and goblet-cell hyperplasia in mice with OVA-induced asthma. No signs of toxicity were observed in mice treated with Cur-PLGA-DPPs. These findings demonstrate the potential of Cur-PLGA-DPPs as a drug-delivery system for lung-targeted asthma treatment.

## Figures and Tables

**Figure 1 pharmaceutics-12-00739-f001:**
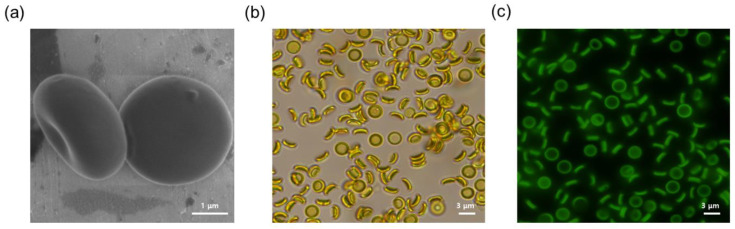
(**a**) Scanning electron micrograph of Cur-PLGA-DPPs. Imaging of Cur-PLGA-DPPs through (**b**) bright-field microscopy, and (**c**) fluorescence microscopy.

**Figure 2 pharmaceutics-12-00739-f002:**
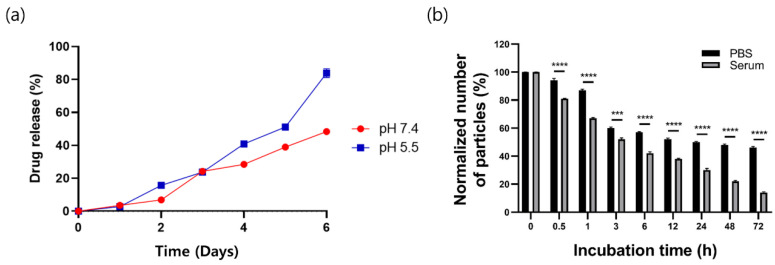
(**a**) Release profile of Cur-PLGA-DPPs in PBS. (**b**) Stability of Cur-PLGA-DPPs in PBS and mouse serum at 37 °C, over different time periods. Values are presented as mean ± standard deviation (asterisks indicate *p*-values; *** *p* < 0.005, and **** *p* < 0.001, compared to data for the 0 h incubation time).

**Figure 3 pharmaceutics-12-00739-f003:**
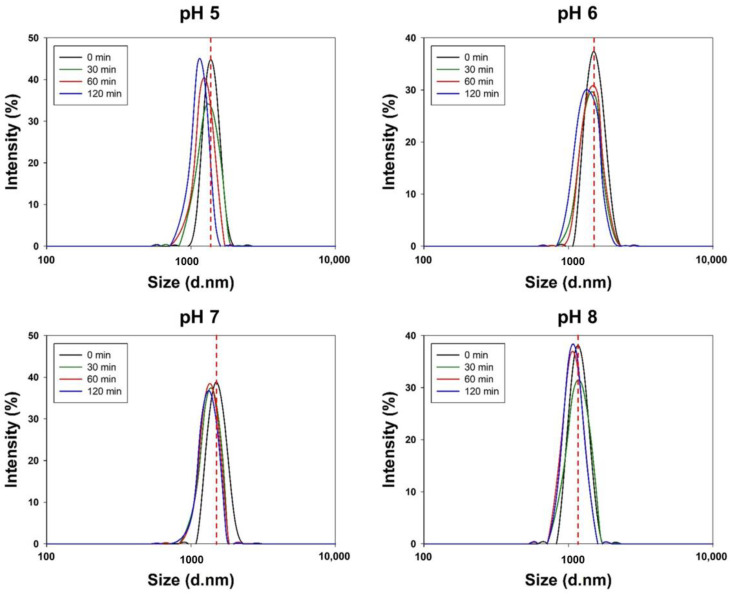
Size distribution of Cur-PLGA-DPPs at different pH values.

**Figure 4 pharmaceutics-12-00739-f004:**
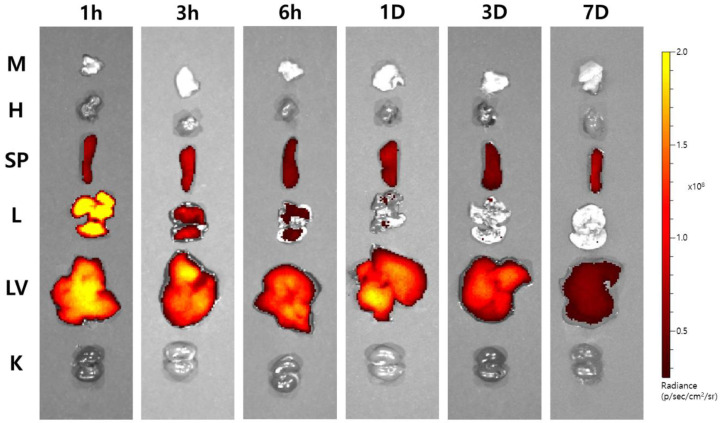
Ex vivo fluorescence images of OVA-induced asthmatic mice at 1 h, 3 h, 6 h, 1 days, 3 days, and 7 days after injection with Cy7-labeled Cur-PLGA-DPPs (*n* = 3 per group). The following organs were examined: muscle (M); heart (H); spleen (SP); lungs (L); liver (LV); kidneys (K). Data are shown in p/sec/cm^2^/sr.

**Figure 5 pharmaceutics-12-00739-f005:**
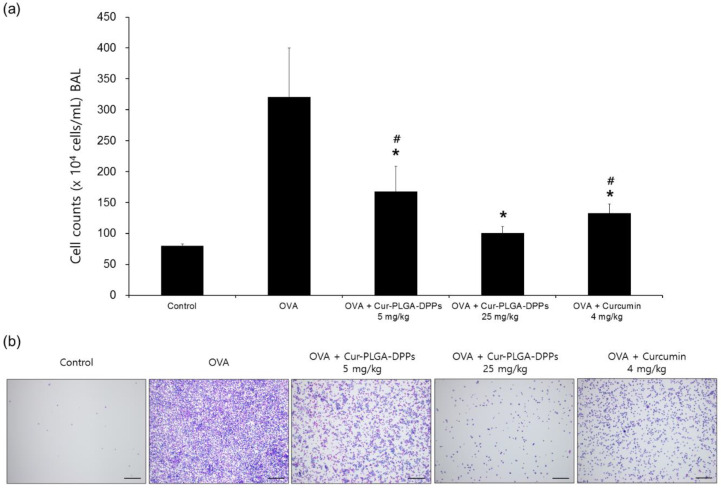
Effect of Cur-PLGA-DPPs on total cells in BALF; (**a**) total cell numbers and (**b**) representative photograph of BALF cells obtained through centrifugation and stained using Diff-Quick. Asterisks indicate a significant difference from the OVA group. Data are shown as mean ± standard deviation (*n* = 4) * *p* < 0.05 compared with the OVA group; ^#^
*p* < 0.05 compared with the OVA + 25 mg/kg Cur-PLGA-DPP-treated group. The scale bars represent 200 µm.

**Figure 6 pharmaceutics-12-00739-f006:**
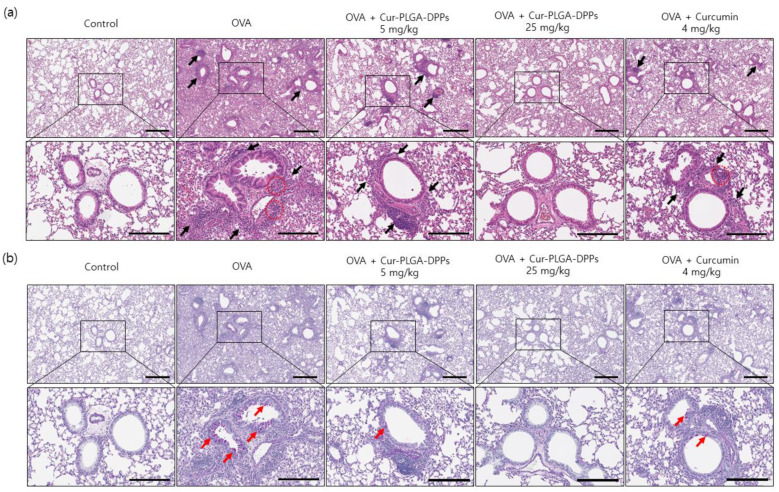
Histological analysis of the effects of Cur-PLGA-DPPs on OVA-induced asthmatic mice. (**a**) Representative results for the H&E staining of lung tissue. Black arrows indicate peribronchial dense inflammatory infiltration; red circles indicate prominent eosinophil infiltration. The scale bars represent 400 µm (upper panel) and 200 µm (lower panel). (**b**) Representative PAS staining of lung tissue. Red arrows indicate increased purple-colored PAS-positive mucin-containing goblet cells (goblet-cell hyperplasia). The scale bars represent 400 µm (upper panel) and 200 µm (lower panel).

**Figure 7 pharmaceutics-12-00739-f007:**
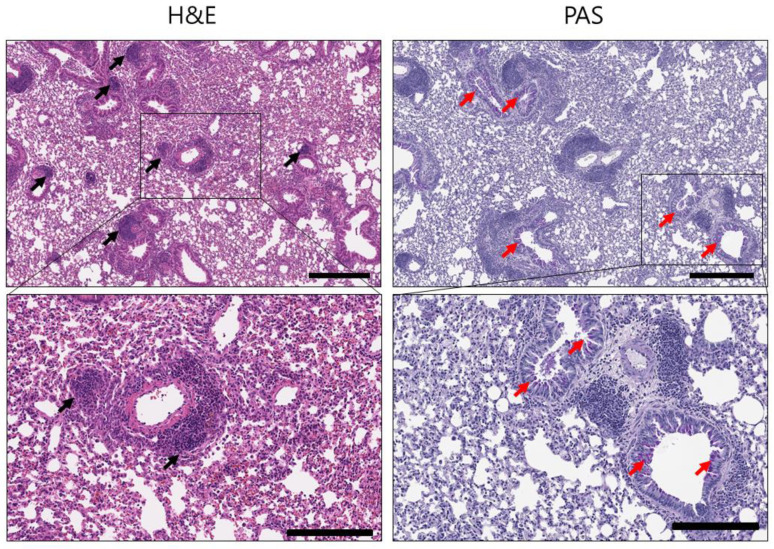
Histological analysis of the effects of blank PLGA-DPPs on OVA-induced asthmatic mice. Representative H&E-stained (**left panel**) and PAS-stained (**right panel**) sections of the lungs. Black arrows in the H&E staining image indicate the infiltration of inflammatory cells. Red arrows in the PAS staining image indicate PAS-positive goblet cells. The scale bars represent 400 µm (**upper panel**) and 200 µm (**lower panel**).

**Table 1 pharmaceutics-12-00739-t001:** Size, zeta potential, and drug-loading capacity of curcumin-containing poly(lactic-*co*-glycolic acid)-based microscale discoidal polymeric particles (Cur-PLGA-DPPs).

Name	Mean Particle Size (μm)	ζ-Potential (mV)	Loading Content (%)
PLGA-DPPs	2.5 ± 0.7	−27.7 ± 4.7	-
Cur-PLGA-DPPs	2.5 ± 0.4	−34.6 ± 4.8	16.3 ± 0.1

**Table 2 pharmaceutics-12-00739-t002:** Effect of administration of Cur-PLGA-DPPs on the biochemical parameters of the liver (ALP, AST, ALT) and kidney (CRE, BUN) damage markers. Values are presented as mean ± standard deviation (*n* = 4).

Parameter	Control	OVA	OVA + Cur-PLGA-DPPs 5 mg/kg	OVA + Cur-PLGA-DPPs 25 mg/kg	OVA + Curcumin 4 mg/kg
ALP (U/L)	362.0 ± 11.3	285.2 ± 36.9	296.8 ± 16.6	294.0 ± 14.3	317.7 ± 26.8
AST (U/L)	107.5 ± 6.36	125.2 ± 53.7	116.0 ± 27.4	115.3 ± 22.2	113.0 ± 9.9
ALT (U/L)	72.0 ± 32.5	57.2 ± 32.9	65.5 ± 34.8	55.8 ± 7.4	48.0 ± 28.3
CRE (mg/dL)	0.2 ± 0.0	0.3 ± 0.1	0.2 ± 0.1	0.2 ± 0.1	0.2 ± 0.1
BUN (mg/dL)	27.4 ± 0.1	26.4 ± 3.5	31.2 ± 3.3	20.3 ± 3.4	23.7 ± 2.8

ALP: alkaline phosphatase, AST: aspartate aminotransferase, ALT: alanine aminotransferase, CRE: creatinine, BUN: blood urea nitrogen.

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
