# Peer review of "Therapeutic Efficacy of Curcumin Enhanced by Microscale Discoidal Polymeric Particles in a Murine Asthma Model"

_pharmaceutics, 2020, doi:10.3390/pharmaceutics12080739_

Round 1

Reviewer 1 Report

The manuscript provides an interesting account of a promising strategy for lung-targeted asthma therapy. But this work must be improved, and some suggestions are:

  1. English grammar should be improved. For example, line 19, “prepare using”; line 60, “However, despite…”
  2. The second paragraph in the Introduction is too long. You cannot put so many information in one paragraph.
  3. Why the particles have to be produced to discoidal? You should introduce its advantages.
  4. Section 3 should be improved. For example, in Section 3.1.3, you only described the experiment result, but should explain why.

Author Response

Reviewer #1:

The manuscript provides an interesting account of a promising strategy for lung-targeted asthma therapy. But this work must be improved, and some suggestions are:

  1. English grammar should be improved. For example, line 19, “prepare using”; line 60, “However, despite…”

Response: Thank you for the suggestion. Based on your advice, the grammatical and proofreading mistakes in the manuscript have been carefully checked and corrected by a professional English editing service.

  1. The second paragraph in the Introduction is too long. You cannot put so many information in one paragraph.

Response: We thank the reviewer for the suggestion. As suggested, we have removed one paragraph (line 50 ~ line 55) from the Introduction section in the revised manuscript.

  1. Why the particles have to be produced to discoidal? You should introduce its advantages.

Response: Thank you for the valuable suggestion. Previous studies demonstrated that discoidal particles tend to accumulate in the lungs about four times more than spherical beads and eight times more than the cylindrical and quasi-hemispherical particles (P. Decuzzi et al. J. Control Release, 2010, 141, 320–327). We have added this information in the Introduction section as follows.

“The size and shape of particles can affect their biological behavior in vivo [22,23]. Decuzzi et al. demonstrated that discoidal particles accumulate in significantly higher concentrations in the lungs than spherical particles or quasi-hemispherical particles [24]. This study performed by them also showed that 3.0 μm-sized particles exhibited significantly higher lung uptake than the 0.7, 1.0, and 2.5-μm sized particles. In addition, in our recent in vivo biodistribution and bioimaging study on 3.0-μm sized discoidal particles, we found that 3 μm-sized discoidal particles were mainly accumulated in lung parenchyma and remained there for up to three days after intravenous administration [25]. The results of these studies suggested the potential of micro-sized discoidal particles for drug delivery to the lungs.”

  1. Section 3 should be improved. For example, in Section 3.1.3, you only described the experiment result, but should explain why.

Response: Thank you for the comment. As per your suggestion, we have added the following information in the Results and Discussion section 3.1.3.

“PLGA polymers are degraded owing to the hydrolysis of the ester bond between lactic acid and glycolic acid [19], and hydrolytic degradation of PLGA can be accelerated under acidic conditions [31].”

Reviewer 2 Report

The authors in work entitled: “Enhanced therapeutic efficacy of curcumin by microscale discoidal polymeric particles in a murine asthma model“ demonstrated interesting results of curcumin loaded particles application in an asthma model. A significant difference was found between curcumin and curcumin loaded particles. The efficacy of asthma model treatment was also significantly improved by curcumin loaded particles. The present manuscript is well written with appropriate discussion of achieved results. I suggest to accept it in the present form.

Author Response

Reviewer #2:

The authors in work entitled: “Enhanced therapeutic efficacy of curcumin by microscale discoidal polymeric particles in a murine asthma model“ demonstrated interesting results of curcumin loaded particles application in an asthma model. A significant difference was found between curcumin and curcumin loaded particles. The efficacy of asthma model treatment was also significantly improved by curcumin loaded particles. The present manuscript is well written with appropriate discussion of achieved results. I suggest to accept it in the present form.

Response: We would like to thank the reviewer for the valuable comments on our manuscript.

Reviewer 3 Report

  1. in the introduction, the authors should explain why the microsized delivery system is better than other drug carriers. The authors might need to cover the main challenges of lung drug delivery.
  2. In figure 2b, the author showed the "normalized particle number" decrease dramatically when incubated with serum. They assume the reason is the degradation caused by the serum components. Is it possible that the particle aggregate, letting the size of the particle fell outside the size range that the machine could test? This can explain why the drug release (2a) is slower than the speed of particle lost in 2b. More importantly, if this is the case, the aggregation might clog the capillaries when administrating the particles in vivo.
  3. In conclusion, the authors conclude the particles mainly present in the lung and liver. However, in figure 4, the amount of fluorescence in the spleen remained constant and high. The author should include spleen in the conclusion. 

Author Response

Reviewer #3:

  1. in the introduction, the authors should explain why the microsized delivery system is better than other drug carriers. The authors might need to cover the main challenges of lung drug delivery.

Response: We thank the reviewer for the valuable suggestions. Previous studies demonstrated that 3.0 μm-sized particles had a greater tendency to accumulate in the lungs than particles measuring 0.7, 1.0, or 2.5 μm (P. Decuzzi et al. J. Control Release, 2010, 141, 320–327). In addition, studies have shown that particles larger than 150 nm generally accumulate in the liver and spleen and those with a diameter less than 20 nm are cleared by renal excretion (C. Dhand et al. RSC Adv. 2014, 4, 32673–32689; E. Blanco et al. Nat. Biotechnol. 2015, 33, 941–951). The typical mouse lung capillary is about 5.7 μm in diameter (A. Geelhaar et al. Respir. Physiol. 1971, 11, 354–366). Thus, we hypothesized that 3 μm-sized discoidal polymeric particles can be accumulated in the precapillary arterioles and capillaries of the lungs. In our previous study, we showed that Cy5.5-labeled 3-μm DPPs were mainly accumulated in the lung parenchyma. They passed through the pulmonary arteries and were trapped in the pulmonary capillary bed, followed by distribution and extravasation into the lung parenchyma. In addition, radioisotope-labeled 3 μm-sized discoidal polymeric particles were accumulated in the lungs at 2 h after injection, and their concentration remained high even on day 1 (J.Y. Park et al. Biomaterials, 2019, 218, 119331). In accordance with the reviewers’ comments, we have added this information in the Introduction section as follows.

“The size and shape of particles can affect their biological behavior in vivo [22,23]. Decuzzi et al. demonstrated that discoidal particles accumulate in significantly higher concentrations in the lungs than spherical particles or quasi-hemispherical particles [24]. This study performed by them also showed that 3.0 μm-sized particles exhibited significantly higher lung uptake than the 0.7, 1.0, and 2.5-μm sized particles. In addition, in our recent in vivo biodistribution and bioimaging study on 3.0-μm sized discoidal particles, we found that 3 μm-sized discoidal particles were mainly accumulated in lung parenchyma and remained there for up to three days after intravenous administration [25]. The results of these studies suggested the potential of micro-sized discoidal particles for drug delivery to the lungs.”

  1. In figure 2b, the author showed the "normalized particle number" decrease dramatically when incubated with serum. They assume the reason is the degradation caused by the serum components. Is it possible that the particle aggregate, letting the size of the particle fell outside the size range that the machine could test? This can explain why the drug release (2a) is slower than the speed of particle lost in 2b. More importantly, if this is the case, the aggregation might clog the capillaries when administrating the particles in vivo.

Response: Thank you for the comment. We assessed the in vitro stability of Cur-PLGA-DPPs in mouse serum at 37 °C over different time periods (Figure 2b). During the experiment, aggregation between particles was not observed in mice serum. The zeta potential value of Cur-PLGA-DPPs was -34.6 ± 4.8 mV. This suggested that Cur-PLGA-DPPs had sufficient dispersion stability. Besides, no death or signs of toxicity were seen in mice dosed intravenously with Cur-PLGA-DPPs in the in vivo cytotoxicity experiment. However, at this point, it is not clear which serum components may cause the degradation of Cur-PLGA-DPPs; thus, further studies would be required.

  1. In conclusion, the authors conclude the particles mainly present in the lung and liver. However, in figure 4, the amount of fluorescence in the spleen remained constant and high. The author should include spleen in conclusion.

Response: We agree with this comment. In the results and discussion section, line 266, we had previously stated: “Cy7-labeled Cur-PLGA-DPPs occurred predominantly in the lungs, liver, and spleen.” As per the reviewer’s comment, the sentence has been rewritten as follows in the conclusion section of the revised manuscript.

“The in vitro release experiment showed sustained release of curcumin from Cur-PLGA-DPPs, and ex vivo biodistribution experiments indicated that Cur-PLGA-DPPs accumulated mainly in the lungs, liver, and spleen.”

Round 2

Reviewer 1 Report

the authors have addressed the concerns I raised in the original review and therefore, recommended the paper be accepted at its current form.